# Dynamic Response Characteristics of Railway Subgrade Using a Newly-Developed Prestressed Reinforcement Structure: Case Study of a Model Test

**DOI:** 10.3390/ma15196651

**Published:** 2022-09-25

**Authors:** Qishu Zhang, Wuming Leng, Junli Dong, Fang Xu

**Affiliations:** 1School of Civil Engineering, Central South University, Changsha 410075, China; 2School of Civil Engineering, Central South University of Forestry and Technology, Changsha 410004, China; 3MOE Key Laboratory of Engineering Structures of Heavy Haul Railway, Central South University, Changsha 410075, China

**Keywords:** prestressed reinforcement structure, scale model test, acceleration response, dynamic performance

## Abstract

Poor subgrade conditions usually induce various subgrade diseases in railways, leading to some adverse influences. An innovative technology that involves installing a prestressed reinforcement structure (PRS) that consists of steel bars and lateral pressure plates (LPP) for subgrade was introduced to improve its stress field and provide compulsive lateral deformation constraints for slope. In this study, an investigation into the dynamic acceleration responses of railway subgrade strengthened according to different PRS schemes was presented using a 1:5 scale model test, aiming to explore the effects of the axle load, the reinforcement pressure, and the loading cycles on the acceleration characteristics of the subgrade. The experimental results showed that (1) after pretension of the steel bar, prestress loss occurred due to the soil creep behavior and group anchor effect, so a moderate amount of over-tension in practices would be necessary; (2) a distinctive periodical behavior of subgrade subjected to the cyclic loads was observed, the horizontal accelerations were generally less than the vertical accelerations at the same measurement heights, and the vibration energy attenuated gradually from the shoulder to the toe along the slope; (3) in the short-term tests, the peak accelerations at all measurement points had a linear correlation with the axle load, and oppositely, it showed an approximately linear decrease with the increasing reinforcement pressure; And (4) in the long-term tests, to simulate the heavy haul wagon with a 35 t axle load, the variation in the effective acceleration with loading cycles under reinforcement pressure 100 kPa initially exhibited a decrease and subsequently tended to be stable, which is apparently less than that without reinforcement pressure. Consequently, it was demonstrated that the PRS itself and increasing reinforcement pressure can effectively mitigate the subgrade vibration, and provide an appropriate alternative to improve the dynamic performance of railway subgrade under the moving train loads.

## 1. Introduction

In recent years, with the rapid development trends of passenger and freight transportation expansion, the current railways in operation cannot match this increasing demand, and faster and heavier trains would be the most direct and effective way to meet these challenges. However, increasing train speeds and axle loads can inevitably lead to an increase in the load frequencies and magnitudes acting on the railway subgrade and accelerate the micro-structural damage of the subgrade. Consequently, a series of possible railroad subgrade diseases, such as unacceptable settlement, subgrade slope instability, cracking in the shoulder section, and ballast pockets would develop and seriously hamper the operational safety and efficiency of existing railways [1,2]. These diseases are closely related to a poor stress state of the subgrade soil and a lack of lateral deformation constraint of the subgrade slope. To this end, a set of newly-developed prestressed reinforcement structure (PRS) consisting of two lateral pressure plates (LPP) and a pretension tendon protected by a sleeve tube was proposed by Leng [3] to strengthen railway subgrade, as shown in Figure 1. Compared with the conventional reinforcement methods for railway subgrades [4], this new structure can not only supplement additional confining pressure for subgrade soil, but also provide lateral deformation constraints for subgrade slope without blocking or disturbing the normal traffic operation.

After years of research, many useful analytical methods associated with prestressed subgrade have been proposed, including investigations into the diffusion characteristics of additional stresses of prestressed embankment in the directions both perpendicular and horizontal to the slope [5,6], the establishment of a simple stability analysis method for prestressed subgrade considering the additional stress propagation effect [7], the technical realization of the PRS in a numerical simulation and further investigation into the mechanical and deformation characteristics of a prestressed railway embankment in the static working state [8], and an optimal design method of the layout spacing for multiple LPPs [9]. These studies have mainly focused on the statics condition, but studies on the dynamic response of prestressed subgrade have not been prominent. Hence, it is imperative to extend the current research topics to capture the dynamic response behaviors of prestressed subgrade.

In terms of the studies on the dynamic characteristics of railway subgrade, a great deal of experimental studies with full-sized physical model tests and field tests have been carried out to comprehensively understand the dynamic properties of railway track–subgrade structure. For example, Bian et al. [10] proposed a prediction approach to determine railway subgrade settlement under the cyclic loading caused by high-speed train loads, which was calibrated using full-scale physical model testing. Kennedy et al. [11] explored the influence of the load level and subgrade stiffness on track settlement for a typical UK track by performing a full-scale laboratory model test and established a track settlement prediction model based on the recorded data. Zhang et al. [12] presented a full-size physical model to study the track settlement of the ballasted railway track–subgrade system under repeated high-speed train loads and proposed a settlement prediction model based on the recorded data. Zhai et al. [13] designed and constructed a full-scale multi-functional test platform for a high-speed railway track–subgrade system and investigated both the dynamic characteristics and long-term performance evolution. Fortunato et al. [14] carried out an assessment of the improvement effects of old railway subgrade reinforced with short soil–binder columns under cyclic loading implemented on full-scale physical models, which indicated that this reinforcement method contributed to a reduction of 15–20% in peak vertical accelerations monitored on the sleepers. Zhang et al. [15] reported a real-size laboratory test of a double-block ballastless track–subgrade system with emphasis on investigating the dynamic performance evolution. In addition, extensive field tests on investigating the dynamic responses of the railway track–subgrade structure and the ground during train operation have been conducted in some studies [16,17,18,19,20,21,22,23,24], but it is not always practicable, since this method is time-consuming, high-cost, and may interrupt train operations.

As is well known, numerical simulation is also a powerful technique to evaluate the mechanical performance of a track–subgrade system. Nevertheless, it generally simulates the real condition with many assumptions and simplifications on the boundary conditions and distribution of the soil properties, and thus leads to difficulty in validating the accuracy and effectiveness of the calculation results, and in situ experimental results are usually required for verification [13,25,26]. Moreover, although it is more persuasive to perform a full-scale model test to investigate the dynamic response of prestressed subgrade, this method is rather time-consuming and expensive as well, and a more advanced and powerful testing platform is required. For these reasons, selecting a reduced-scale model test as an effective solution is preferable. Certainly, abundant experimental studies on railroad dynamic performance have also been conducted using laboratory-scaled models. For instance, Brown et al. [27] developed a new laboratory scaled model test facility, where the dynamic stresses of subgrade, the transient deflection of the sleeper, and the permanent track settlement of a ballasted railway subjected to moving train loads were investigated. Shaer et al. [28] explored the dynamic performance of a ballasted railway by conducting a reduced-scale experiment equipped with three sleepers under dynamic loading. Ishikawa et al. [29] experimentally evaluated the dynamic behavior of the track–subgrade system of a single-line ballasted railway under repeated train loads using a 1:5 scale model test. Momoya et al. [30] revealed the deformation characteristics of a ballasted railway embankment at the structural transition zone under a moving train load with a 1:5 reduced-scale model. Chen et al. [31] designed a 1:4 scale test model of the double-line track–subgrade structure of the high-speed ballastless railway and found that in the subgrade bottom layer, the attenuation of dynamic stress was faster under both larger axle loads and higher frequency conditions.

In railroad infrastructures, the subgrade plays a critical part in the track stiffness and degradation of the track geometry [32]. The vibration acceleration response analysis is generally employed as an important detection index to assess the dynamic stability of the subgrade under repeated train loads [30,31,33,34]. Therefore, it is also necessary to understand the effect of PRS on the subgrade dynamic acceleration response and the long-term behavior under moving train loads, which is helpful for the optimization design of prestressed subgrade.

The existing studies on conventional railway track–subgrade structure provide some valuable references for understanding the dynamic response characteristics of this new prestressed subgrade. However, prestressed subgrade, as a new subgrade structure, is notably different from conventional railway subgrade, which makes these existing studies unable to be directly applied in the revelation of the dynamic properties of prestressed subgrade and the evaluation of the effect of PRS on the dynamic response behavior of subgrade. In this regard, a laboratory dynamic test model system for prestressed subgrade with a scale ratio of 1:5 was developed, based on which a series of dynamic loading tests were performed to investigate the effects of the axle load, pretension force (or reinforcement pressure), and loading cycle (including short-term and long-term cyclic loads) on subgrade outline accelerations in both the vertical and horizontal directions. The obtained results can provide a reference for the subsequent theoretical study on the dynamic behaviors of prestressed subgrade and play a guiding role in the design and construction of prestressed subgrade.

## 2. Experimental Overview of Physical Model Test

In this section, similitude principles, the model geometry and testing system composition, the materials and boundary treatment used, the subgrade construction and quality detection, the test program, and the sensor arrangement are described.

### 2.1. Similitude Principle

Dynamic responses generated in a reduced-scale system should be consistent with an actual prototype, but complete similarity between a prototype and a model that can make an exact prediction of the prototype behavior is difficult to satisfy in reality [35,36]. The Buckingham π theorem [37] serves as an important tool for dimensional analysis, and it can simplify the physical issue by considering the dimensional homogeneity, thus contributing to the reduction of the related variable numbers. Given the principle of stress field similarity, the actual laboratory conditions, difficulty in preparing subgrade fillings, and time costs, this study selected the geometric size *L*, time *T*, and mass *M* as a consistent fundamental dimension system, and set the geometric size *L*, elastic modulus *E*, and density *ρ* as control variables with scale factors of *C_L_* = 5, *C_E_* = 1 and *C_ρ_* = 1, respectively. The main similitude parameters required in this model test are summarized in Table 1 based on the Buckingham π theorem [37,38], and the detailed deduction process related to these physical variables is described below.

Physical function relationships among these physical variables *Q_i_* should be satisfied as
(1)$$f\left(Q_{i}\right)=f\left(L,E,\rho ,\sigma ,t,m,P,s,a,f,\varphi ,\mu ,c\right)=0$$
while each physical variable can be expressed by the three fundamental dimensions as
(2)$$\left[Q_{i}\right]=M^{k_{1}}\cdot L^{k_{2}}\cdot T^{k_{3}}$$
where the exponents *k*_1_, *k*_2_, and *k*_3_ denote dimensionless numbers; and *Q_i_* represents the list of 13 variables involved in Equation (1).

Totally, there were 13 variables involved in this study, and three control dimensions were included. Thus, the remaining 10 independent dimensionless groups π*_i_* should be determined in the form of:(3)$$\pi_{i}=Q_{i}\cdot \left(L^{x_{1}}\cdot E^{x_{2}}\cdot \rho^{x_{3}}\right)$$

Subsequently, with the inclusion of the dimensional expressions all of physical quantities listed in Table 1 into Equation (3), it can be rewritten as
(4)$$\pi_{i}=Q_{i}\cdot {\left(L\right)}^{x_{1}}\cdot {\left(ML^{-1}T^{-2}\right)}^{x_{2}}\cdot {\left(ML^{-3}\right)}^{x_{3}}$$

Based on the requirements of dimension homogeneity, each *π_i_* has a general relationship of the form as
(5)$$\begin{array}{ll} \left[1\right]=M^{0}L^{0}T^{0} & =M^{x_{2}+x_{3}}\cdot L^{x_{1}-x_{2}-3x_{3}}\cdot T^{-2x_{2}}\cdot Q_{i}\left(M,L,T\right)\\ & =M^{k_{1}+x_{2}+x_{3}}\cdot L^{k_{2}+x_{1}-x_{2}-3x_{3}}\cdot T^{k_{3}-2x_{2}} \end{array}$$

In additional, the exponents regarding *M*, *L* and *T* must meet the following system equations:(6)$$\left\{ \begin{array}{l} M:0=k_{1}+x_{2}+x_{3};\\ L:0=k_{2}+x_{1}-x_{2}-3x_{3};\\ T:0=k_{3}-2x_{3} \end{array}\right.$$

By performing this method repeatedly, all the π*_i_* groups can be derived by the dimension matrix method, and finally, the scale factors of all variables can be obtained as listed in Table 1.

### 2.2. Model Geometry and Testing System Composition

The laboratory model test systems are composed of different parts, as shown in Figure 2. To construct the subgrade and install the PRS conveniently, a large-scale U-shaped rigid model box with internal net dimensions of 2.9 m long, 3.86 m wide, and 1.6 m high to reduce the box effect [39], whose natural frequency is far from that of the subgrade soil, was manufactured using the No. 16 channel beam. To further reduce or avoid the dynamic boundary effect induced by the model box on the subgrade, 10 cm-thick polystyrene foam sheets were laid along the bottom and around the inner wall of the model box. For the subgrade part, the subgrade slope gradient was adjusted from the conventional 1:1.5 to 1:1.0 to better exhibit the advantages of PRS, and a standard single-line heavy haul railway embankment prototype in accordance with the Chinese Code for Design of Heavy Haul Railway (TB 10625-2017) [40] was adopted to design the structural geometry dimensions of the subgrade model. Specifically, the subgrade top and the cross-section of the foundation were 1.66 m and 3.66 m wide, respectively; the thicknesses of the ballast layer, the subgrade bed layer, the body layer, and the foundation were 0.1 m, 0.5 m, 0.5 m, and 0.2 m, respectively; the longitudinal length of the subgrade was 2.7 m. The cross-sectional and longitudinal layouts and detailed dimensions are presented in Figure 3, in which the *x*-axis denotes the horizontal direction of the subgrade cross-section, the *y*-axis represents the longitudinal direction of the line, and the *z*-axis is the vertical direction. Additionally, in Figure 2 and Figure 3, the *R_i_* is the *i*th row of the LPPs, while *C_i_* is the *i*th column of the LPPs.

There were 25 LPPs made of C40 reinforced concrete with a bottom size of 28 cm × 28 cm, which were placed in the middle region at each side of the slope and occupied a longitudinal length of 1.4 m. Correspondingly, each steel bar with a diameter of 12 mm was equipped with a PVC protective pipe with an inner diameter of 14.2 mm to separate the steel bars from the subgrade soils, and all of them were horizontally embedded in the subgrade at different heights. The nuts were used to anchor the pretension of the steel bars that were tensioned by a hollow jack, and the ends of the steel bar were connected with the LPPs on both sides of the slope. Meanwhile, the tendon dynamometers were employed as a practical tool to monitor and adjust the pretension and installed between the nut and LPP, as shown in Figure 2 and Figure 3.

For the track structure (i.e., rail, sleeper, and ballast components), five sleepers were designed in this system, and in the model, No. 10 steel I-beams with a length of 0.52 m were used to replace the Ⅲa-type prestressed concrete sleepers in the prototype, and their flange width was trimmed from 6.8 cm to 6.4 cm to match the similitude rules. In addition, the No. 10 steel I-beams 0.60 m in length were also used to substitute for the rails with complex cross-sections in the prototype. The spacings of the sleepers and rails were 0.12 m and 0.287 m, respectively. Note that the contact edges between the rails and sleepers were welded.

### 2.3. Track-Subgrade-Foundation Construction

The ballast layer in the model was filled with a 0.1 m-thick graded macadam mixture, the particle size of which was mainly in the range of 5~15 mm so that its uniformity could be ensured, and hence, the layout of the track components would be more convenient.

To simulate the natural ground condition, a foundation layer 0.2 m thick was constructed at the model bottom, on which the subgrade, consisting of a subgrade bed and body layers, was laid, and the entire model was constructed by layer. For the fillers in the model, machine-made sand and two low-liquid-limit clays were mixed at different mass ratios to prepare the fillers for these three layers. According to the soil classification system provided by the Chinese Code for Design of Railway Earth Structure (TB 10001-2016) [41], the filler for the subgrade bed layer can be classified as a well-graded fine breccia with the soil of Group A2, and two other different fine breccia soils of Group B3 were constructed for the subgrade body and foundation layers, respectively. The particle size distributions of these three fillers are plotted in Figure 4. The basic physical properties of the fillers are listed in Table 2.

As highlighted in Figure 5, the major construction processes in the model making were as follows: (a) drawing reference lines for each compacted layer and the slope lines; (b) preparation of the fillers; (c) leveling of the loose fillers; (d) dynamic compaction; (e) installation of fence plates to avoid the lateral spread of the filler during the dynamic compaction; (f) embedding steel bars and their PVC protective sleeve tubes; (g) dismantling the fence plates; (h) slope cutting; (i) drawing marker lines for small steps; (j) excavation of the small steps; (k) installation of the LPPs; (l) installation of the tendon dynamometer and nuts. An electric compactor with a 300 mm by 300 mm vibrating plate was employed to compact each filler layer to 10 cm thick. After the compaction of each layer, a conventional spirit level was adopted to roughly check the layer’s flatness. Meanwhile, a *K*30 plate load test using a spot-check method and a compaction test for each layer were performed as the key control indexes to ensure the compaction quality, and one group of the detection results is exhibited in Figure 6. It is evident that the construction quality met the relevant requirements.

### 2.4. Loading Platform and Test Program

The loading platform, as shown in Figure 2, consisted of a reaction frame, distribution beam, and one servo-hydraulic actuator above the rails. The maximum output force and frequency were 50 kN and 5 Hz, respectively, for the actuator. The distribution beam, employing a combination of two 3 cm-thick steel plates welded at the flange ends of one Q235 H-shaped steel with dimensions of 60 cm × 25 cm × 25 cm, equally allocated the dynamic load to two rails and then transmitted such a load to the five sleepers below.

Compared with ordinary/high-speed railways, the operation speeds of trains on heavy haul railways are relatively lower, but a higher dynamic load is applied on the track–subgrade system due to the heavier axle load. Thus, enough attention should be paid to the dynamic load amplitude rather than loading frequency (i.e., the operation speed) in the model test because the dynamic load induced by heavy haul transportation exerts more significant influence on the dynamic response degree and the long-term accumulative deformation of the subgrade. To efficiently obtain a train-induced dynamic load imposed on the sleeper, a mathematical solution, the Euler-Bernoulli elastic beam (E-BEB) theory [42], can be used to calculate the contact force between a one-sided rail and a sleeper in a ballasted railway due to its simplicity and reasonably accurate prediction [43,44]. Therefore, it was an effectively equivalent approach to simulate the moving train loads. First, the wheel load considering the dynamic effect can be formulated as follows:(7)$$P_{d}=P_{0}\left(1+\alpha v\right)$$
where *P*_0_ is the static axle load (kN), and *α* is the speed influence factor, generally equal to 0.0045 when the train speed *v* is lower than 120 km/h [45].

The reaction force *R* provided by one end of the sleeper to balance the force allocated from a one-wheel load acting on the rail can be formulated by the following equation [46]:(8)$$R=\frac{P_{d}ka}{2}\sum_{i=1}^{n}\varphi \left(x_{i}+vt\right)$$
where *k* is the track characteristic parameter (m^−1^), *k* = (*u*/4*EI*)^1/4^; *a* is the sleeper spacing (m); *u* is the track foundation modulus (kN/m^2^), and *u* = *D*/*a*, where *D* is the rail support stiffness (kN/m); *E* and *I* are the elastic modulus (kN/m^2^) and the moment of inertia of the rail (m^4^), respectively; *n* denotes the number of wheels or axles considered; *x_i_* represents the distance between the wheel *i* and the target sleeper at a particular time *t*; finally, the distribution function can be expressed as *φ*(*x*) = e^−*k*|x|^(cos*k*x + sin*k*|x|).

In this model test, five sleepers were designed to share the dynamic loads acting on the rails. This is because moving train loads applied on rails are generally undertaken by approximately five sleepers (fasteners) [47,48], and the time-varying loads at each sleeper could be calculated based on the prototype track structure dimensions and related similitude relations. Due to the limitation of only one actuator in the experimental loading facility, a resultant load by superposing these five loading histories with the consideration of the time phase lag (Δ*t* = *a*/*v*) between adjacent sleepers was applied on the distribution beam. An example of the M-shaped load-time history, which was derived from a freight wagon with a 350 kN axle load traveling at 80 km/h based on the above-mentioned approach, is illustrated in Figure 7, where the peak load was 37.8 kN and the loading frequency was 8 Hz.

As mentioned previously, considering the limitation of loading facility capacity in lab, a distiAs mentioned previously, considering the limitation of the loading facility capacity in a lab, a distinct difference between the input and the real output waveforms could be observed when the loading frequency was set to 8 Hz in the preliminary tentative test. For this reason, the loading frequency was decreased to *f* = 2 Hz in all subsequent loading tests so as to ensure the loading process with the desired accuracy.

The test program can be divided into two series: (a) short-term loading tests and (b) long-term loading tests. In the former series, five pretension forces (i.e., reinforcement pressures were 0 kPa, 25 kPa, 50 kPa, 75 kPa, and 100 kPa) and five axle loads (230 kN, 250 kN, 270 kN, 300 kN, and 350 kN) were considered, and each case performed 2000 loading cycles. Note that the pretension forces in the first row of steel bars were merely tensioned to half of that of the other rows, avoiding the local shear failure near the shoulder. For the latter, the heaviest axle loads of 350 kN were adopted to simulate the worst case, and two extreme reinforcement cases, in which 0 kPa and 100 kPa were applied, were implemented as a pronounced comparison to reflect the reinforcement effect of the PRS. Apart from that, it is worth noting that the sine wave was employed as the input waveform in the long-term tests, rather than an M-shaped wave. This was mainly due to the fact that the impact effect of the M-shaped wave made the loading position difficult to sustain its stability, thus resulting in the loading devices gradually and slightly deviating from the initial position during the long-term testing process. However, the loading amplitude and frequency regarding the two loading waves were kept at the same level. The dynamic test schemes are detailed in Table 3.

### 2.5. Sensor Arrangement

The acceleration responses of the prestressed subgrade were measured using an 891-Ⅱ accelerometer with a sensitivity of 100 mV/m/s^2^, developed by the Institute of Engineering Mechanics, China Earthquake Administration, Harbin, China. During the test, the middle column (C3) of the LPPs was selected as the measurement section, where three vertical (points V1, V2, and V3) and four horizontal (points H1, H2, H3, and H4) accelerometers were arranged, as shown in Figure 8. Except for the accelerometers installed at points V1 and H1, which were close to the shoulder on the subgrade surface, they were all positioned on the corresponding LPP surfaces at each side of the subgrade slope using wedge-shaped plates, which were glued onto the LPP surface.

## 3. Prestress Loss of Steel Bars

The level of the pretension force of the steel bar (or reinforcement pressure provided by the PRS) is a crucial control factor for a prestressed subgrade, which plays a decisive role in the effectiveness of the PRS. Prior to the implementation of the dynamic loading tests, a preliminary test regarding the evolution of the pretension force of each steel bar was performed by taking the *R*1 = 50 kPa, *R*2 = *R*3 = *R*4 = *R*5 = 100 kPa (i.e., reinforcement case of 100 kPa) as the typical study, with a duration of 74 h after installation of the PRS, briefly capturing the static variation characteristics of the prestress loss. As the pretension forces of each row were anchored, they could be equivalently transformed to the initial reinforcement pressures applied on the subgrade slopes for each row of the PRS. Figure 9 shows the variation in the pretension force over time at the *R*2 row. It was found that the pretension force of each bar had a decreasing tendency over time and the loss magnitudes in sequence were as follows: *C*3*R*2 (0.92 kN) > *C*4*R*2 (0.86 kN) > *C*2*R*2 (0.79 kN) > *C*5*R*2 (0.68 kN) > *C*1*R*2 (0.62 kN). Meanwhile, the attenuation of the pretension force tended mostly to be marginal after 74 h. These observations were largely owing to the soil creep behavior caused by additional stress and the group anchor effect with a nonuniform distribution of additional stress.

The average prestress loss ratios of the steel bars for each row are illustrated in Figure 10. It can be observed that the prestress loss ranged from 6.08% to 13.77% corresponding to the borders of the yellow rectangular, and each row of prestress losses showed an inverted “V” distribution from the *C*1 to the *C*5 column separating with the red dash line, with the largest and least prestress losses of the steel bars occurring at the middle column (i.e., column *C*3) and both ends (i.e., column *C*1 or *C*5), respectively, which possibly resulted from the distribution characteristics of the additional stress provided by the PRS. In addition, judging from each row, the average prestress loss ratios were as follows: *R*3 (11.96%) > *R*4 (10.91%) > *R*2 (9.62%) > *R*5 (8.57%) > *R*1 (7.52%). These findings imply that in practical engineering, it is necessary to provide an over-tension treatment for the steel bars during the tension of the steel bars. Indeed, further investigation into the prestress loss perhaps caused by soil creep, the moving train load, and the climate environment is necessary.

## 4. Results and Discussions of Short-Term Loading Tests

### 4.1. Acceleration Response at Measurement Points

Figure 11 shows the variation in the vertical acceleration over time during the stable loading phase in the case of an axle load of 270 kN and reinforcement pressure of 50 kPa, indicating that the accelerometer can accurately record the repetition process of loading–unloading excited by the actuator. In addition, it is apparent that at the time instant of reaching the peak load, the acceleration at each measuring point also rapidly increased to the peak level and were subsequently accompanied by some fluctuations. The acceleration histories are basically symmetrical with respect to the baseline of 0 m/s^2^. A distinct periodicity with a time interval of 0.5 s can be found in these recorded responses. The average values of the peak accelerations gradually decreased along the slope from the shoulder to the toe in a descending sequence, as follows: 0.1353 m/s^2^ (V1) > 0.1284 m/s^2^ (V2) > 0.1220 m/s^2^ (V3). This is mainly attributed to the material damping and diffusion effect of the dynamic loads, resulting in a gradual attenuation of vibration downward along the subgrade slope.

Figure 12 presents the horizontal acceleration–time histories at each measurement point in the case of a 270 kN axle load and 50 kPa of reinforcement pressure. One can see that these response curves are also symmetrical with 0 m/s^2^, and the distributions of the horizontal accelerations down the slope are similar to those of the vertical acceleration responses, but with different magnitudes. However, in terms of the variation degree of these time-varying acceleration histories, the horizontal acceleration response seems to have been more intensified than the vertical counterpart. The magnitudes of the peak accelerations at the four measurement points were as follows: 0.1013 m/s^2^ (H1) > 0.0995 m/s^2^ (H2) > 0.0893 m/s^2^ (H3) > 0.0412 m/s^2^ (H4). For the measurement points at the same height, the horizontal accelerations were generally lower than the vertical accelerations, but the former reached approximately 75% of that of the latter, which indicates that the train-induced horizontal vibration cannot be ignored.

It is also confirmed that during the application of repeated loads, the accelerations in both directions in other cases had similar variation shapes to the case mentioned above (i.e., axle load: 270 kN; reinforcement pressure: 50 kPa), but with different magnitudes. The peak accelerations at seven measurement points in the 25 test cases are summarized in Table 4.

### 4.2. Influence of Axle Load

The magnitude of the axle load varied from 230 kN to 350 kN, with the aim to explore its influence on the acceleration response of the subgrade. Figure 13 shows the variations in the accelerations in both directions with an axle load at three reinforcement pressures of 0 kPa (a), 50 kPa (b), and 100 kPa (c). The peak accelerations in both the horizontal and vertical directions increased as the axle load increased, exhibiting an approximately linear relationship. In detail, the incremental range of vertical peak accelerations was up to 60.3~74.4% as the train axle load increased from 230 kN to 350 kN, while the increment in the horizontal direction was merely in the range of 26.3~35.0%, implying that the vertical acceleration was more significantly influenced by the train axles than the horizontal acceleration.

### 4.3. Influence of Reinforcement Pressure

The reinforcement pressure varied from 0 kPa to 100 kPa to capture its influence on the peak accelerations, as illustrated in Figure 14. The distribution of the acceleration with reinforcement pressures under three axle loads of 230 kN, 270 kN, and 350 kN is presented. It can be seen that the accelerations in both directions showed a nearly linear decrease with an increase in the reinforcement pressure. As shown in Figure 14b, under a 270 kN axle load, the vertical and horizontal accelerations of the subgrade at a reinforcement pressure of 100 kPa were lower than that at 0 kPa, and the corresponding variation ranges were 89.17~91.34% and 91.7~96.15%, respectively. Therefore, in terms of the reinforcement effect, although the PRS was unable to significantly mitigate the vibration acceleration response in the short-term tests, a visibly positive effect provided by the PRS on the subgrade dynamic stability could still be observed. Certainly, there was an upper limit of the reinforcement pressure for the subgrade to avoid local shear failure, and it is suggested that the shear strength and compressibility of subgrade soil should be tested before designing PRS schemes, providing a reasonable basis for the determination of the optimal reinforcement pressure for the practical engineering.

Based on all the short-term experimental results, the reinforcement mechanism of prestressed subgrade can be deduced as follows: as the train passes, the vertical and lateral deformations were generated on the subgrade top surface and slope, respectively. Meanwhile, with the help of the PRS in the prestressed subgrade, the slope lateral deformation was restricted because the LPPs placed on both sides of the slope are tensioned by a steel tendon with pretension. Specifically, since Young’s modulus of the steel tendon is greatly higher than that of the subgrade soil, a PRS with high tensile rigidity (*EA*) together with the subgrade soil itself could resist the lateral deformation and share the outward lateral thrust force induced by the upper dynamic load when subjected to moving train loads, which contributes to an improvement in the lateral deformation resistance for subgrade. The reinforcement principle of prestressed subgrade is demonstrated in Figure 15, where the red arrows refer to the reinforcement pressure, the yellow arrows represent the loading transfer path, and the white arrows show the tensile resistance effect of steel bars. In addition, given that the subgrade could be regarded as a deformation continuum medium, once the development of the slope lateral deformation is partly inhibited by the PRS, the subgrade settlement would also be reduced accordingly. On the other hand, it is clear that the vertical strain development of the soil unit would be limited when its lateral pressures in the horizontal direction are raised, which is in accordance with the generalized Hooke’s law. That is, improving the confining pressure of the subgrade could decrease the vertical deformation and also weaken the vibration response in the prestressed subgrade. These descriptions above could reveal the mechanical principle and working advantages of prestressed subgrade well.

## 5. Results and Discussions of Long-Term Loading Tests

The long-term dynamic response is a crucial index to evaluate the service performance of railway subgrade under repetitive cyclic loads. As aforementioned, due to the limitation of the test facility, the manufacturing accuracy of the steel components, the flatness of the ballast surface, the impact property of the M-shaped wave, and the component positions of the rails and sleepers were unable to stay in the initial position for a long period during the long-term tests. Therefore, a sine wave (at a 2 Hz frequency, with 37.8 kN simulating the heaviest axle load of 350 kN), with the same peak and loading frequency as the M-shaped wave, was adopted as the input load in the long-term tests, as presented in Figure 16, and only two reinforcement pressures of 0 kPa and 100 kPa were applied in the long-term tests for direct comparison purposes.

The peak acceleration refers to the maximum vibration response of the subgrade, which corresponds to the worst conditions at the time instant, but it is unsuitable to directly evaluate the long-term vibration. Here, a total loading time length of 15,000 s was needed for each of the test cases, as shown in Figure 17, and the real vibration effect would be evaluated inaccurately if all the peak accelerations in each loading cycle were considered alone. Thus, the calculation of the running root mean square root (RMS) value with respect to the acceleration–time history was used as an effective evaluation method to obtain the acceleration characteristics in the time domain more reasonably, as given by [23]:(9)$$a_{\mathrm{RMS}}\left(i\right)=\sqrt{\frac{1}{t_{0}}\int_{0}^{t_{0}}a^{2}\left(t\right)dt}$$
where *t*_0_ is the width of the time window, and *t*_0_ = 10 s in this study; *a*_RMS_(*i*) refers to the effective RMS value of acceleration for the *i*th time window.

Figure 18 presents the RMS of the acceleration evolution over time at two reinforcement pressures of 0 kPa and 100 kPa at point V1, giving an obvious comparison of the reinforcement effect with and without reinforcement pressure. This indicates that in the case of 0 kPa, the RMS of the acceleration history was generally kept at a relatively steady state with some fluctuations; whereas, in the case of 100 kPa, the RMS of the acceleration curve exhibited a decreasing tendency during the initial loading phase, which may have been an adjustment process due to the interaction between the PRS and the subgrade soil subjected to 100 kPa of reinforcement pressure, followed by a gradual stable trend until the end of the test. In the first 2500 cycles, the effective RMS values at the two reinforcement pressures were in the range of 0.0455~0.0462 m/s^2^ and 0.0377~0.039 m/s^2^, respectively, and the latter was about 0.836 times that of the former. When the two curves developed to a stable phase, the average RMS values of acceleration in the cases of 0 kPa and 100 kPa approximated 0.0467 m/s^2^ and 0.0377 m/s^2^, respectively, and the latter was merely 80.9 percent of the former. These observations indicate that the vibration response induced by a moving train load will attenuate with an increase in the reinforcement pressure provided by the PRS. Furthermore, in practical engineering, the optimal design of a reinforcement scheme with the PRS for railway subgrade could effectively regulate its dynamic response to a certain degree.

## 6. Conclusions

Prestressed subgrade is a newly developed railway subgrade structure or reinforcement method for conventional subgrade via improving its stress state and forming compulsive lateral constraints on both sides of the slope. This study aimed to investigate the effect of the axle load, reinforcement pressure, and loading cycle on the train-induced vibration acceleration through an indoor reduced-scale dynamic model test. Based on the results obtained, the main conclusions can be drawn as follows:(1)After the pretension of the steel bar, there was a prestress loss due to the soil creep behavior and anchor group effect. Specifically, after 74 h of pretension, the prestress loss of the steel bars in each row was around 10% on average. Consequently, it is suggested to provide moderate over-tension for steel bars in practice.(2)An apparent periodic response of the prestressed subgrade subjected to cyclic loads was observed, and the vibration energy gradually attenuated downward of the slope. At the same measurement height, the horizontal peak acceleration was up to 75% of the vertical peak magnitude, implying that the horizontal vibration acceleration of the subgrade induced by the moving train load cannot be neglected.(3)The peak accelerations in both the horizontal and vertical directions showed an approximately linear increase with an increasing train axle load at all of the measurement points, and the influence degree in the vertical direction was relatively more sensitive. The peak acceleration response weakened with an increase in the reinforcement pressure, which means that the increasing reinforcement pressure could contribute to the mitigation of the subgrade vibration, and thus, the smoothness and safety of train operations can be improved. However, the reinforcement pressure had an upper limit considering both the shear strength and compressibility of the subgrade soil, and thus, it is necessary to further investigate the optimal reinforcement pressure for various subgrade conditions.(4)In the long-term tests, without reinforcement pressure, the effective value of the vertical acceleration of the subgrade was almost unchanged with the increasing cyclic loads; in contrast, once the reinforcement pressure reached 100 kPa, it showed a decreasing tendency first and then tended to be stable as the loading cycles increased, and the latter value was around 81% of the former.

## Figures and Tables

**Figure 1 materials-15-06651-f001:**
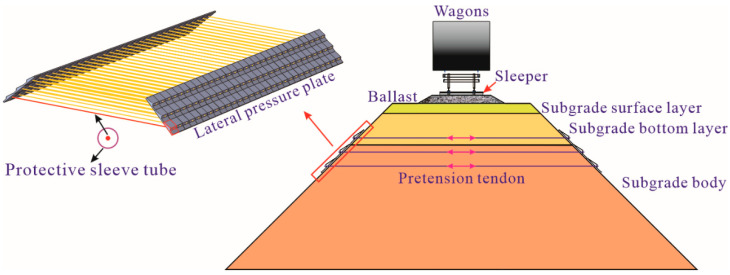
Diagram of prestressed railway subgrade.

**Figure 2 materials-15-06651-f002:**
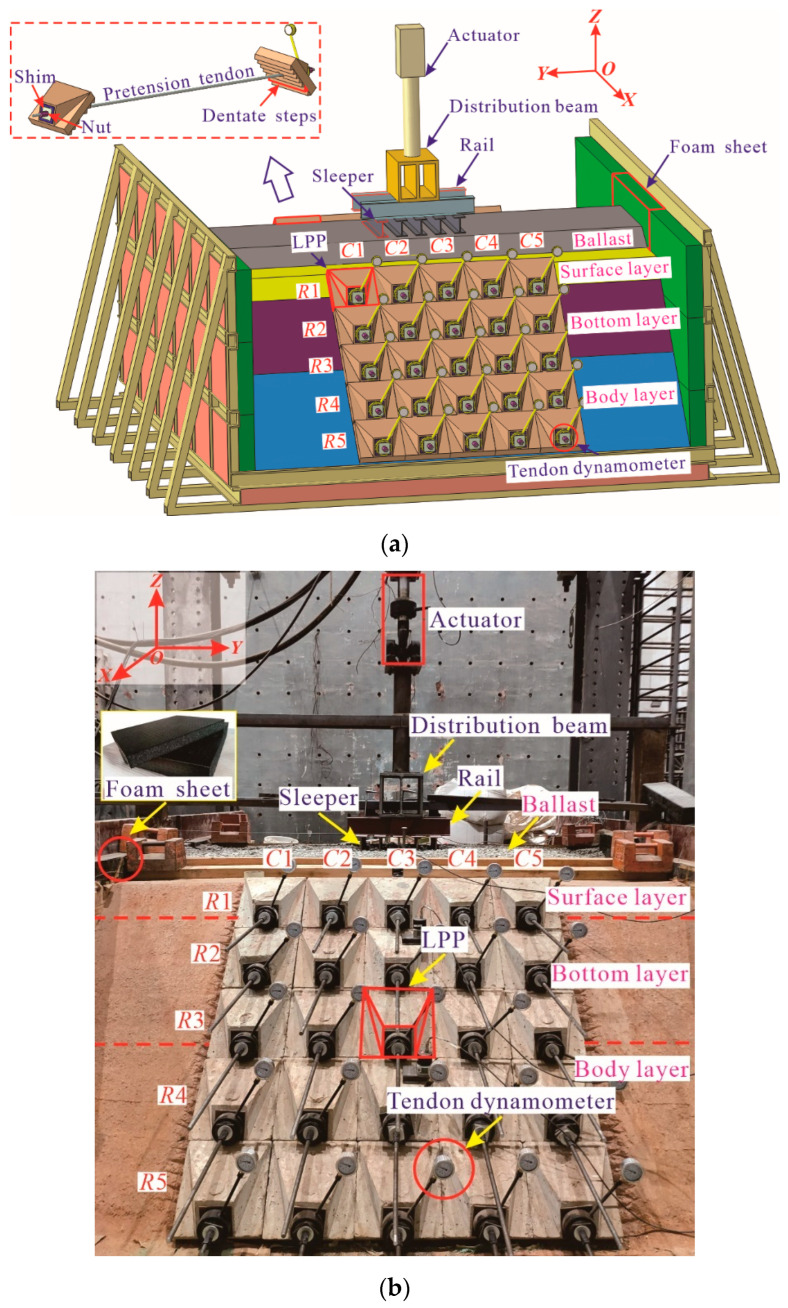
Test model system of prestressed subgrade. (**a**) Schematic diagram; (**b**) physical photograph.

**Figure 3 materials-15-06651-f003:**
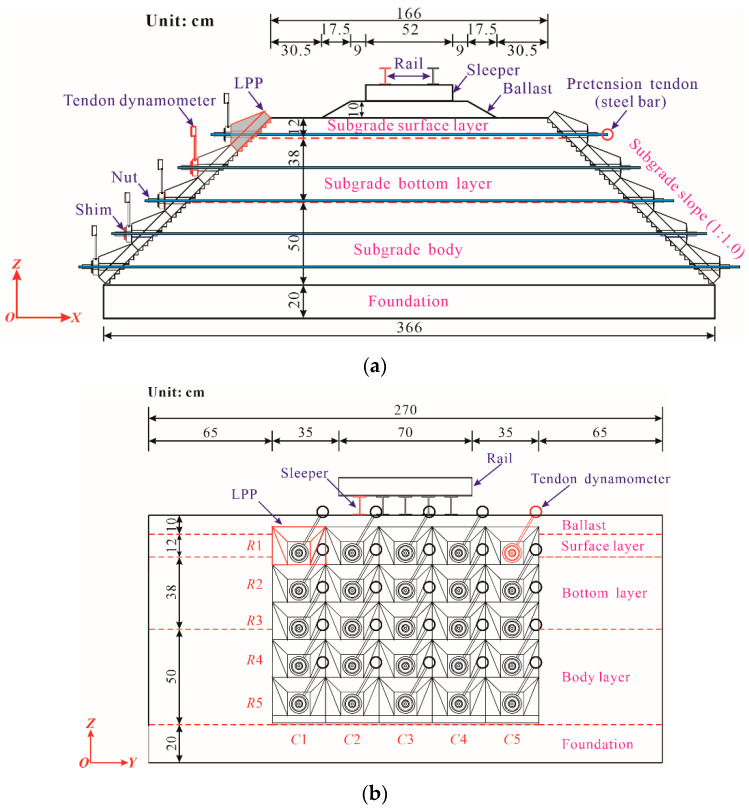
Model dimensions of the prestressed subgrade. (**a**) Cross-sectional layout; (**b**) longitudinal layout.

**Figure 4 materials-15-06651-f004:**
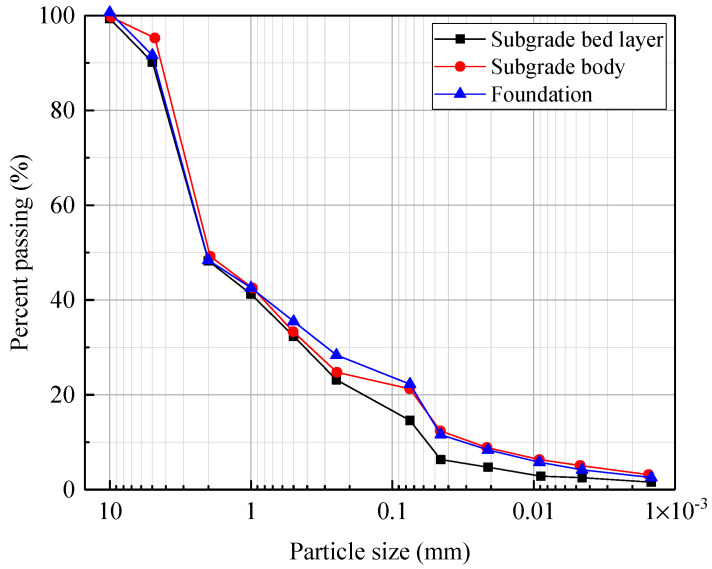
Particle size distribution of fillers.

**Figure 5 materials-15-06651-f005:**
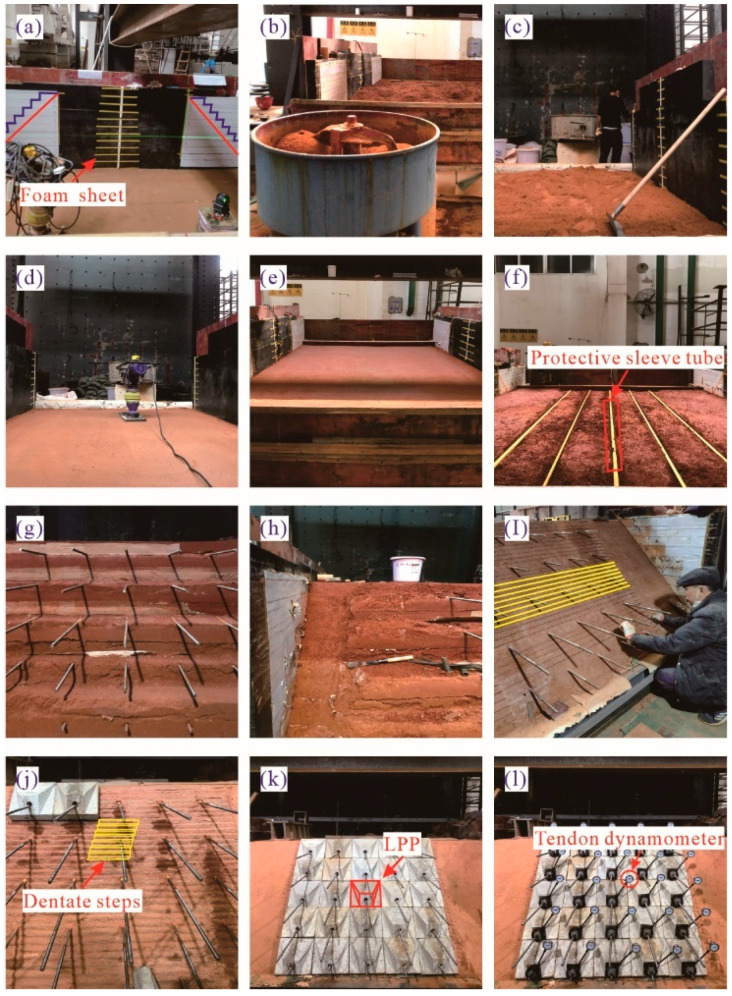
Construction process of prestressed subgrade.

**Figure 6 materials-15-06651-f006:**
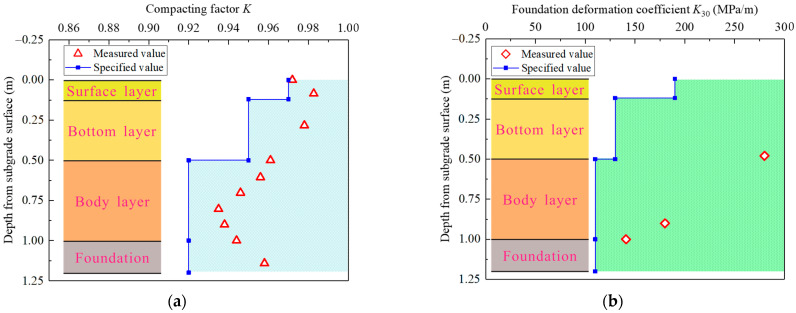
Detection results of compaction quality of subgrade and foundation. (**a**) Compacting factor *K*; (**b**) foundation deformation coefficient *K*_30_.

**Figure 7 materials-15-06651-f007:**
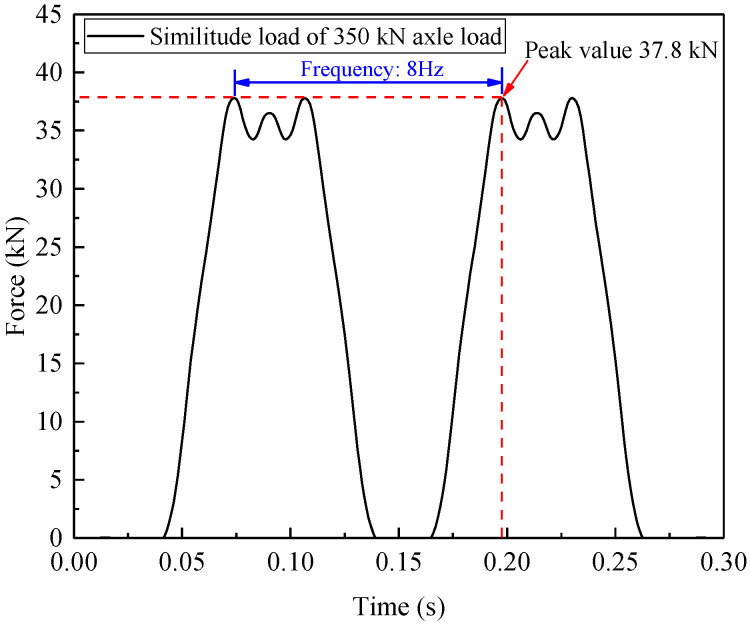
Target input load of the actuator to equivalently simulate a 350 kN axle load at 80 km/h.

**Figure 8 materials-15-06651-f008:**
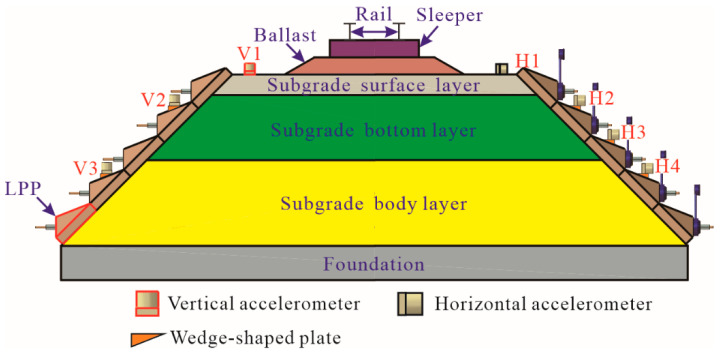
Arrangement of acceleration measurement points at column C3.

**Figure 9 materials-15-06651-f009:**
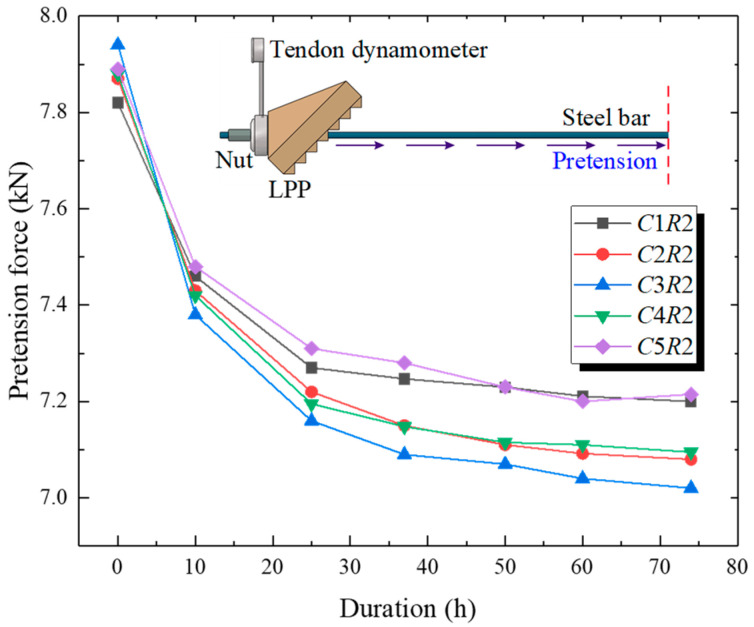
Variation in pretension force over time at row *R*2.

**Figure 10 materials-15-06651-f010:**
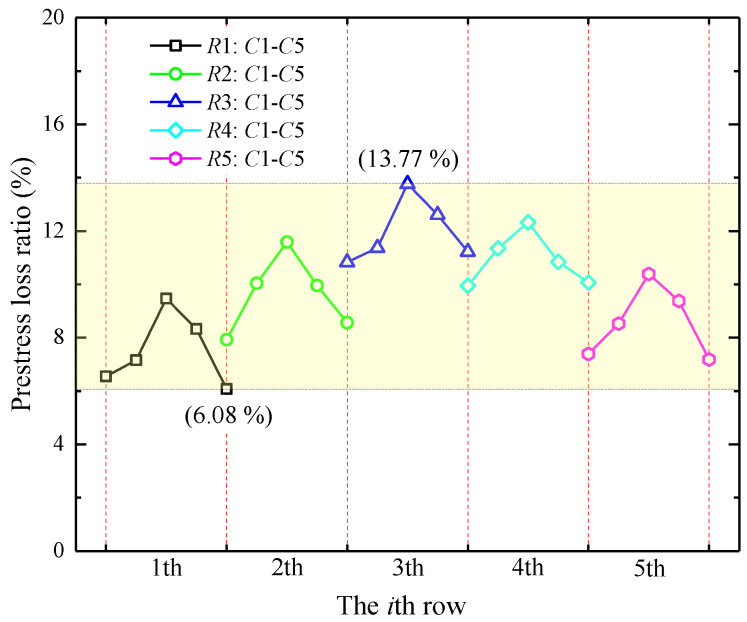
Prestress loss ratio of steel bars.

**Figure 11 materials-15-06651-f011:**
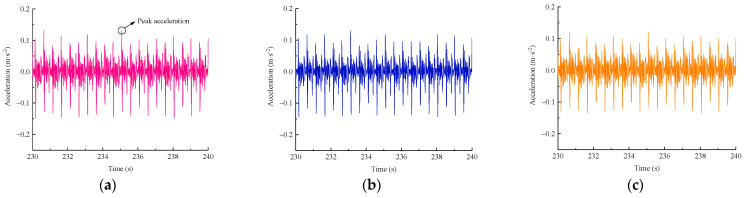
Vertical acceleration–time histories. (**a**) Point V1; (**b**) Point V2; (**c**) Point V3.

**Figure 12 materials-15-06651-f012:**
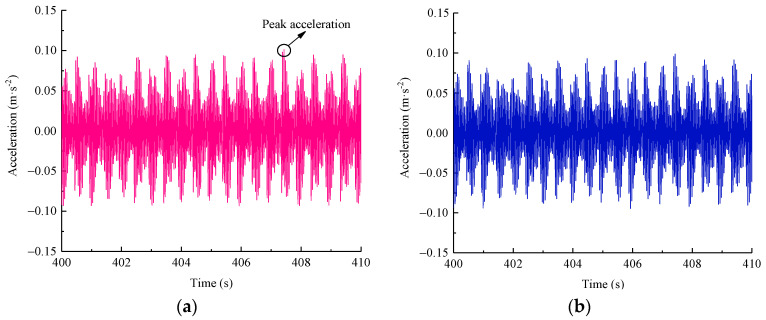
Horizontal acceleration–time histories. (**a**) Point H1; (**b**) Point H2; (**c**) Point H3; (**d**) Point H4.

**Figure 13 materials-15-06651-f013:**
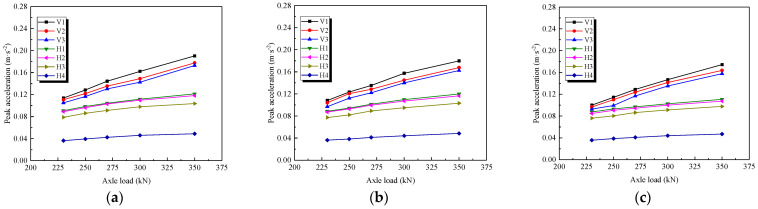
Relationship between peak acceleration and axle load at different reinforcement pressures. (**a**) Reinforced pressure 0 kPa; (**b**) Reinforced pressure 50 kPa; (**c**) Reinforced pressure 100 kPa.

**Figure 14 materials-15-06651-f014:**
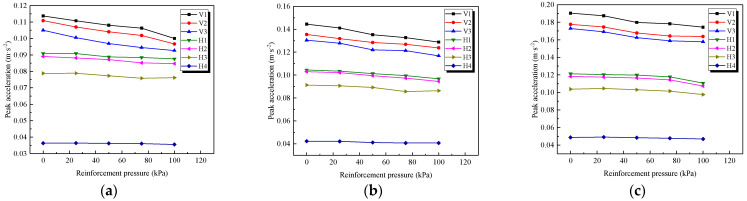
Variations in peak accelerations with reinforcement pressures under different axle loads. (**a**) 230 kN axle load; (**b**) 270 kN axle load; (**c**) 350 kN axle load.

**Figure 15 materials-15-06651-f015:**
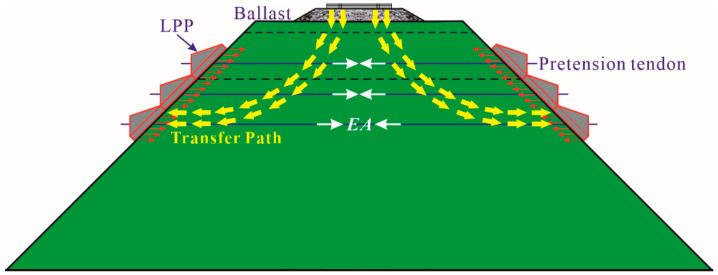
Schematic diagram of the mechanical principle of prestressed subgrade.

**Figure 16 materials-15-06651-f016:**
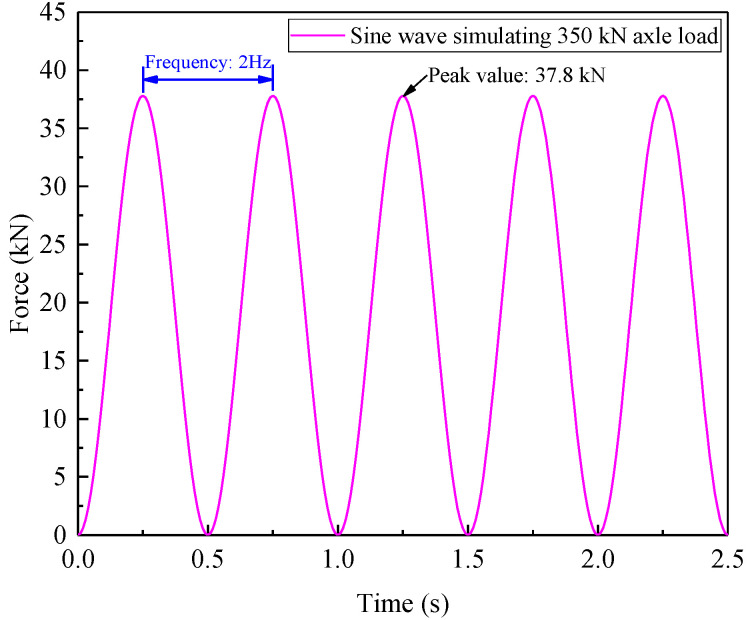
Loading history of sine wave simulating a 350 kN axle load.

**Figure 17 materials-15-06651-f017:**
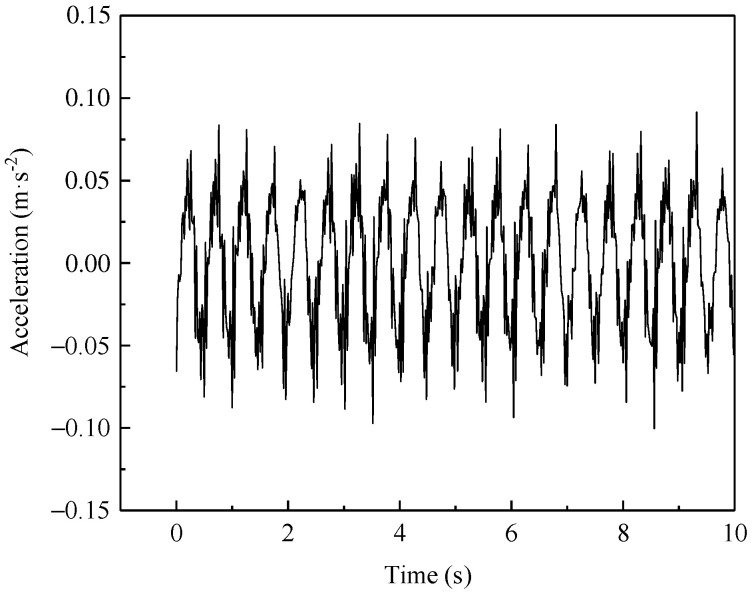
Measured vertical acceleration–time history at measuring point V1.

**Figure 18 materials-15-06651-f018:**
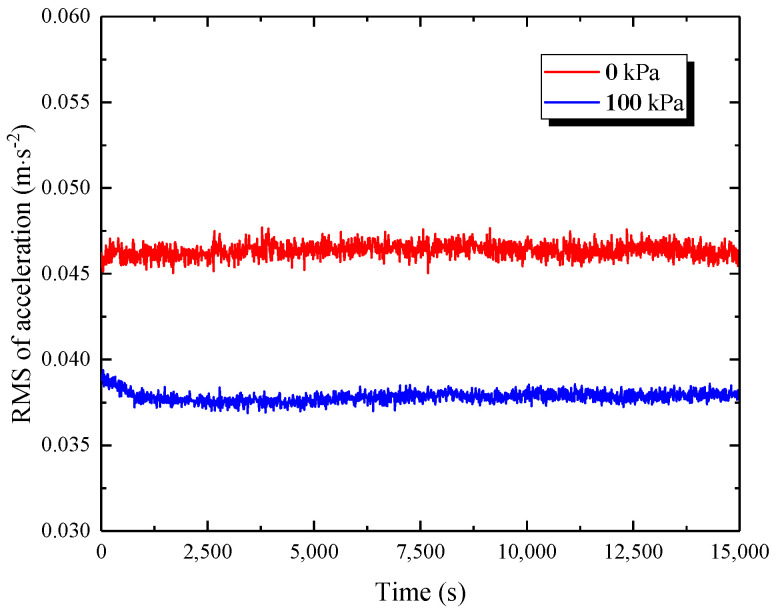
Comparison of the effective RMS variations in two reinforcement cases at point V1.

**Table 1 materials-15-06651-t001:** Scale factors for the main variables in model test.

No.	Variable	Symbol	Dimensions	Deduction	Similarity Ratio (Model/Prototype)
1	Geometric size	*L*	*L*	*C_L_* = *C_L_*	1/5
2	Elastic modulus	*E*	*ML* ^−1^ *T* ^−2^	*C_E_* = *C_E_*	1
3	Density	*ρ*	*ML* ^−3^	*C_ρ_* = *C_ρ_*	1
4	Stress	*σ*	*ML* ^−1^ *T* ^−2^	*C_σ_* = *C_E_*	1
5	Time	*t*	*T*	*C_t_* = (*C_L_*) (*C_ρ_*)^1/2^ (*C_E_*)^−1/2^	1/5
6	Mass	*m*	*M*	*C_m_* = (*C_L_*)^3^ (*C_ρ_*)	1/125
7	Force	*P*	*MLT* ^−2^	*C_P_* = (*C_L_*)^2^ *C_ρ_*	1/25
8	Displacement	*s*	*L*	*C_s_* = *C_L_*	1/5
9	Acceleration	*a*	*LT* ^−2^	*C_a_* = (*C_L_*)^−1^ (*C_ρ_*)^−1^ *C_E_*	5/1
10	Frequency	*f*	*T* ^−1^	*C_f_* = (*C_L_*)^−1^ (*C_ρ_)*^−1/2^ (*C_E_*)^1/2^	5/1
11	Frictional angle	*φ*	*—*	*C_φ_* = 1	1
12	Poisson’s ratio	*μ*	*—*	*C_μ_* = 1	1
13	Cohesion	*c*	*ML* ^−1^ *T* ^−2^	*C_c_* = *C_E_*	1

**Table 2 materials-15-06651-t002:** Basic physical properties of fillers.

Layer	Maximum Dry Density *ρ*_dmax_ (kg/m^3^)	Optimum Moisture Content *ω*_opt_ (%)	Uniformity Coefficient (*C*_u_)	Coefficient of Gradation (*C*_c_)	Fillers Group
Subgrade bed	2192	6.80	45.59	1.19	A2
Subgrade body	2187	6.41	89.51	2.26	B3
Foundation	2173	7.17	83.02	1.08	B3

**Table 3 materials-15-06651-t003:** Details of dynamic model tests.

No.	Test Series	Reinforcement Pressure (kPa)	Loading Cycles	Axle Load (kN)	Loading Wave
1	Short-term	0	2000	230	M-shaped wave
2		25		250	
3		50		270	
4		75		300	
5		100		350	
6	Long-term	0	30,000	350	sine wave
7		100			

**Table 4 materials-15-06651-t004:** Peak accelerations in all test cases.

Axle Load (kN)	Reinforcement Pressure (kPa)	Peak Accelerations at All Measurement Points (m/s^2^)						
		V1	V2	V3	H1	H2	H3	H4
230	0	0.1137	0.1108	0.1050	0.0909	0.0890	0.0788	0.0364
	25	0.1107	0.1069	0.1005	0.0908	0.0882	0.0788	0.0364
	50	0.1080	0.1040	0.0969	0.0888	0.0872	0.0773	0.0363
	75	0.1062	0.1018	0.0944	0.0884	0.0852	0.0758	0.0361
	100	0.1000	0.0967	0.0927	0.0876	0.0847	0.0761	0.0356
250	0	0.1284	0.1218	0.1165	0.0984	0.0962	0.0861	0.0395
	25	0.1238	0.1203	0.1134	0.0977	0.0954	0.0851	0.0395
	50	0.1234	0.1206	0.1123	0.0943	0.0928	0.0820	0.0383
	75	0.1212	0.1186	0.1065	0.0932	0.0915	0.0814	0.0392
	100	0.1144	0.1100	0.0994	0.0930	0.0905	0.0803	0.0387
270	0	0.1445	0.1355	0.1304	0.1045	0.1030	0.0913	0.0424
	25	0.1411	0.1318	0.1278	0.1035	0.1022	0.0907	0.0422
	50	0.1353	0.1284	0.1220	0.1013	0.0995	0.0893	0.0412
	75	0.1327	0.1269	0.1213	0.0995	0.0973	0.0857	0.0408
	100	0.1288	0.1237	0.1168	0.0968	0.0944	0.0864	0.0407
300	0	0.1622	0.1491	0.1425	0.1118	0.1098	0.0979	0.0460
	25	0.1615	0.1510	0.1441	0.1107	0.1084	0.0971	0.0454
	50	0.1574	0.1449	0.1400	0.1097	0.1071	0.0949	0.0441
	75	0.1478	0.1417	0.1337	0.1059	0.1035	0.0926	0.0429
	100	0.1468	0.1417	0.1352	0.1025	0.1001	0.0913	0.0439
350	0	0.1903	0.1776	0.1729	0.1213	0.1182	0.1038	0.0488
	25	0.1874	0.1746	0.1691	0.1204	0.1175	0.1046	0.0492
	50	0.1798	0.1677	0.1625	0.1198	0.1165	0.1031	0.0484
	75	0.1782	0.1643	0.1588	0.1177	0.1143	0.1015	0.0478
	100	0.1744	0.1638	0.1578	0.1106	0.1073	0.0976	0.0469

## Data Availability

The data used to support the findings of this study are available from the corresponding author upon reasonable request.

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
