# Peer review of "Dynamic Response Characteristics of Railway Subgrade Using a Newly-Developed Prestressed Reinforcement Structure: Case Study of a Model Test"

_materials, 2022, doi:10.3390/ma15196651_

Round 1

Reviewer 1 Report

The manuscript reports an innovative technology by installing prestressed reinforcement structure with the steel bars and lateral pressure plates. Detailed indoor studies and tests are presented providing the results of investigations on the effect of axle load, reinforcement pressure and loading cycle on the train-induced vibration acceleration. The results reported represent a notable advance in improvement of the dynamic performance of railway subgrade under the moving train loads.

The title and abstract are appropriate for the content of the text. In abstract, the authors summarized the main research question and key findings. The description of study subject and results are detailed and correct. In Introduction, the authors provide a comprehensive summary of previous research and important publications related to this specific topic. Furthermore, the manuscript is well constructed and experiments and tests are well performed. Data collected are interpreted accurately and the results support the conclusion. The manuscript reads well. The figures are clear and readable and support the findings.

Overall, this is a clear, concise, and well-written manuscript that has implications for theoretical basis, design and construction of prestressed subgrade. In my opinion, the manuscript is suitable for publication in Materials in a present form.

Author Response

Point 1: The manuscript reports an innovative technology by installing prestressed reinforcement structure with the steel bars and lateral pressure plates. Detailed indoor studies and tests are presented providing the results of investigations on the effect of axle load, reinforcement pressure and loading cycle on the train-induced vibration acceleration. The results reported represent a notable advance in improvement of the dynamic performance of railway subgrade under the moving train loads.

The title and abstract are appropriate for the content of the text. In abstract, the authors summarized the main research question and key findings. The description of study subject and results are detailed and correct. In Introduction, the authors provide a comprehensive summary of previous research and important publications related to this specific topic. Furthermore, the manuscript is well constructed and experiments and tests are well performed. Data collected are interpreted accurately and the results support the conclusion. The manuscript reads well. The figures are clear and readable and support the findings.

Overall, this is a clear, concise, and well-written manuscript that has implications for theoretical basis, design and construction of prestressed subgrade. In my opinion, the manuscript is suitable for publication in Materials in a present form.

Response 1: Thanks very much for the reviewer’s recognition of this manuscript.

Reviewer 2 Report

A few comments :

- table 2 : please put dry density in the units of the international system

- figure 5 : the caption is too big, it would be better to put the detail of each photo in a paragraph

- figure 11 :  I don't understand why the signal is traced for such a long time, it makes an unusable black rectangle. Zooming in a shorter time is sufficient.

- line 374 : "0 (a), 50 (b) and 100 kN (c)"

- figure 16 represents the plot of the loading for a period of 2.5s. Why does Figure 17 not represent the response over the same period? Which would still avoid an unusable rectangle.

- It would be better to to give the axle loads in newtons in all the paper

- text of reference 1 is shifted (line 488).

Author Response

Response to Reviewer 2 Comments

A few comments :

Point 1:  table 2: please put dry density in the units of the international system.

Response 1: Thanks very much for the reviewer’s valuable comment. As suggested, the unit of the dry density has been revised using the international system of units (SI) with the form of kg/m3. Please see the Table 2 in the revised manuscript, which corresponds to Page 8, Line 243.

Table 2. Basic physical properties of fillers.

Layer

Maximum dry density ρdmax (kg/m3 )

Optimum moisture content ωopt (%)

Uniformity coefficient (Cu)

Coefficient of gradation (Cc)

Fillers group

Subgrade bed

2192

6.80

45.59

1.19

A2

Subgrade body

2187

6.41

89.51

2.26

B3

Foundation

2173

7.17

83.02

1.08

B3

Point 2: figure 5: the caption is too big, it would be better to put the detail of each photo in a paragraph.

Response 2: Thanks very much for the reviewer’s valuable comment. As suggested, the description of the model construction process has been presented by a paragraph. Please see the Lines 228-234 in the revised manuscript.

Lines 228-234:

As highlighted in Fig. 5, the major construction processes in the model making were as follows: (a) drawing reference lines for each compacted layer and the slope lines; (b) preparation of the fillers; (c) leveling of the loose fillers; (d) dynamic compaction; (e) installation of fence plates to avoid the lateral spread of the filler during the dynamic compaction; (f) embedding steel bars and their PVC protective sleeve tubes; (g) dismantling the fence plates; (h) slope cutting; (i) drawing marker lines for small steps; (j) excavation of the small steps; (k) installation of the LPPs; (l) installation of the tendon dynamometer and nuts.

Point 3: figure 11: I don't understand why the signal is traced for such a long time, it makes an unusable black rectangle. Zooming in a shorter time is sufficient.

Response 3: Thanks very much for the reviewer’s valuable comment. As suggested by the reviewer, the acceleration response curves in Fig. 11 have been revised by a shorter time.

Point 4: line 374: "0 (a), 50 (b) and 100 kN (c)".

Response 4: Thanks very much for the reviewer’s valuable comment. We have supplemented the corresponding number of figure behind each reinforcement pressure based on the reviewer’s suggestion. Please see the Lines 390-391 in the revised manuscript.

Lines 390-391:

accelerations in both directions with an axle load at three reinforcement pressures of 0 kPa (a), 50 kPa (b), and 100 kPa (c).

Point 5: figure 16 represents the plot of the loading for a period of 2.5 s. Why does Figure 17 not represent the response over the same period? Which would still avoid an unusable rectangle.

Response 5: Thanks very much for the reviewer’s valuable comment. To avoid an unusable rectangle, the response time in Fig. 17 has been reduced to 10 s (the length of one time window). Please see Fig. 17 in the revised manuscript.

Point 6: It would be better to to give the axle loads in newtons in all the paper.

Response 6: Thanks very much for the reviewer’s valuable comment. As suggested, the unit of the axle loads in the whole manuscript have been modified using the form of “kN”.

Point 7: text of reference 1 is shifted (line 488).

Response 7: Thanks very much for the reviewer’s careful inspection and valuable comment. The format modification of the reference 1 has been made correctly. Please see the Lines 516-517 in the reference section.

Reviewer 3 Report

In this paper, the authors investigated the influence of prestressed reinforced structure (PRS) on the vibration acceleration along vertical and horizontal direction, using downsized railway and train-imitating load. They found that PRS worked to suppress vibration acceleration depending on the increase of reinforcement pressure. The dependence of reinforcement pressure and axle load on the vibration behavior was investigated by short-term test, and they also performed long-term experiment for practical applications. This manuscript is well-documented, and I think just one thing should be addressed before publication.

Is there the optimal reinforcement pressure. When the authors tested 0-100 kPa, peak acceleration decreased linearly as shown in Fig.14. What is estimated to happen over 100kPa for reinforcement pressure?

Author Response

Response to Reviewer 3 Comments

Point 1: In this paper, the authors investigated the influence of prestressed reinforced structure (PRS) on the vibration acceleration along vertical and horizontal direction, using downsized railway and train-imitating load. They found that PRS worked to suppress vibration acceleration depending on the increase of reinforcement pressure. The dependence of reinforcement pressure and axle load on the vibration behavior was investigated by short-term test, and they also performed long-term experiment for practical applications. This manuscript is well-documented, and I think just one thing should be addressed before publication.

Is there the optimal reinforcement pressure. When the authors tested 0-100 kPa, peak acceleration decreased linearly as shown in Fig.14. What is estimated to happen over 100 kPa for reinforcement pressure?

Response 1: Thanks very much for the reviewer’s support and encouragement. There should be a critical reinforcement pressure or an optimal reinforcement pressure based on the experimental results and actual observations in lab. The peak acceleration decreases linearly with the increasing reinforcement pressures within 0-100 kPa, while the plastic yield and even the local shear failure of soil beneath the LPP bottom that results in the functional failure of PRS may be generated as the reinforcement pressure increases continuously. Thereby, it is suggested that the shear strength and compressibility of subgrade soil, especially close to the LPP bottom, should be evaluated using the relevant test methods first in the practical application, and subsequently an optimal reinforcement pressure could be determined to make the subgrade work in the elastic state. Indeed, a deeper investigation on the optimal reinforcement pressure available for different subgrade conditions is desirable. This has been clarified in both Section 4.3 and the conclusion, please see the Lines 407-411 and Lines 496-499 in the revised manuscript.

Lines 407-411:

Certainly, there was an upper limit of the reinforcement pressure for the subgrade to avoid local shear failure, and it is suggested that the shear strength and compressibility of subgrade soil should be tested before designing PRS schemes, providing a reasonable basis for the determination of the optimal reinforcement pressure for the practical engineering.

Lines 496-499:

However, the reinforcement pressure had an upper limit considering both the shear strength and compressibility of the subgrade soil, and thus, it is necessary to further investigate the optimal reinforcement pressure for various subgrade conditions.